# Development and Validation of Risk Prediction Models for Colorectal Cancer in Patients with Symptoms

**DOI:** 10.3390/jpm13071065

**Published:** 2023-06-29

**Authors:** Wei Xu, Ines Mesa-Eguiagaray, Theresa Kirkpatrick, Jennifer Devlin, Stephanie Brogan, Patricia Turner, Chloe Macdonald, Michelle Thornton, Xiaomeng Zhang, Yazhou He, Xue Li, Maria Timofeeva, Susan Farrington, Farhat Din, Malcolm Dunlop, Evropi Theodoratou

**Affiliations:** 1Centre for Global Health, Usher Institute, University of Edinburgh, Edinburgh EH8 9AG, UK; 2Colon Cancer Genetics Group, Medical Research Council Human Genetics Unit, Medical Research Council, Institute of Genetics and Cancer, University of Edinburgh, Edinburgh EH4 2XU, UK; 3Edinburgh Cancer Research Centre, Institute of Genetics and Cancer, University of Edinburgh, Edinburgh EH4 2XU, UK; 4Clinical Research Team, Oncology Department, Forth Valley Royal Hospital, Stirling Road, Larbert FK5 4WR, UK; 5University Hospital Wishaw & University Hospital Monklands, NHS Lanarkshire, Airdrie ML6 0JS, UK; 6Wishaw General Hospital, Wishaw ML2 0DP, UK; 7Danish Institute for Advanced Study, Research Unit of Epidemiology, Biostatistics and Biodemography, Institute of Public Health, University of Southern Denmark, 5230 Odense M, Denmark

**Keywords:** colorectal cancer, symptoms, prediction model, polygenic risk score

## Abstract

We aimed to develop and validate prediction models incorporating demographics, clinical features, and a weighted genetic risk score (wGRS) for individual prediction of colorectal cancer (CRC) risk in patients with gastroenterological symptoms. Prediction models were developed with internal validation [CRC Cases: n = 1686/Controls: n = 963]. Candidate predictors included age, sex, BMI, wGRS, family history, and symptoms (changes in bowel habits, rectal bleeding, weight loss, anaemia, abdominal pain). The baseline model included all the non-genetic predictors. Models A (baseline model + wGRS) and B (baseline model) were developed based on LASSO regression to select predictors. Models C (baseline model + wGRS) and D (baseline model) were built using all variables. Models’ calibration and discrimination were evaluated through the Hosmer-Lemeshow test (calibration curves were plotted) and C-statistics (corrected based on 1000 bootstrapping). The models’ prediction performance was: model A (corrected C-statistic = 0.765); model B (corrected C-statistic = 0.753); model C (corrected C-statistic = 0.764); and model D (corrected C-statistic = 0.752). Models A and C, that integrated wGRS with demographic and clinical predictors, had a statistically significant improved prediction performance. Our findings suggest that future application of genetic predictors holds significant promise, which could enhance CRC risk prediction. Therefore, further investigation through model external validation and clinical impact is merited.

## 1. Introduction

Colorectal cancer (CRC) was the third most common cancer and the second leading cause of cancer-related death in the world, 2022 [1]. Early CRC diagnosis and timely treatment could improve survival. Survival rate depends on cancer stage at diagnosis, with 5-year net survival starting at approximately 90% for stage I and reduced to 10% for stage IV [2]. Although screening has successfully reduced CRC incidence and mortality, the majority of CRCs are still diagnosed after symptomatic presentation [3]. It is important to develop accurate prediction models to identify symptomatic patients with higher CRC risk in whom referral is most appropriate. These models could assist clinical professionals in their decision-making for further clinical care, such as risk-tailored cancer screening, testing, and treatments [4]. 

We have identified 19 prediction models that have been developed for CRC in patients with symptoms [5,6,7,8,9,10,11,12,13,14,15,16,17,18,19,20,21,22]. However, these models used predictors such as basic demographic characteristics (age, sex, BMI), lifestyle factors (smoking, alcohol consumption), biomarkers (haemoglobin, CEA), and clinical features (bowel symptoms). None of them use genetic predictors associated with CRC common susceptibility variants (neither single nucleotide polymorphisms nor polygenic risk scores). Therefore, we aimed to examine the association between a constellation of demographic factors, clinical features, and genetic risk scores in patients with gastrointestinal symptoms and CRC risk. Furthermore, we aimed to develop and to validate prediction models that incorporate significant predictors, enabling personalized prediction of CRC risk in patients with symptoms.

## 2. Materials and Methods

### 2.1. Studies and Variables

CRC prediction models were developed with internal validation in a study that included participants from the Study of Colorectal Cancer in Scotland (SOCCS) (n = 1649) and the Lothian Bowel Symptoms Study (LABSS) (n = 1000). SOCCS, a case-control study, started in 1999 and has been recruiting CRC incident cases (aged ≥ 16 years old) and healthy controls (matched on age, sex, and health board) from across Scotland. In the current study, we only used data from colorectal cancer cases that had developed gastrointestinal symptoms prior to their recruitment in SOCCS. LABSS, which is a multi-centre case-control study started in 2017, recruited patients (aged ≥ 18 years old) with gastrointestinal symptoms through endoscopy, CT scanning, colorectal surgery, and gastroenterology units within NHS recruiting centres across Scotland. SOCCS and LABSS collected age, sex, BMI, family history, and symptoms (changes in bowel habits, rectal bleeding, weight loss, anaemia, abdominal pain). Age (years old), sex (male/female), BMI (kg/m^2^), and family history of CRC (yes/no) were collected and documented in questionnaires by the study nurse in SOCCS and LABSS. We designated individuals as having a positive family history (yes) if their first-degree (e.g., parents, siblings, and children) or second-degree (e.g., grandparent/grandchild, half-siblings, aunt/uncle, and niece/nephew) or any other relatives have a documented history of CRC. In SOCCS, symptoms (yes/no) were collected by the study nurse through GP referral and/or consultant clinic referral letters, as documented in medical records in TRAK (the NHS Lothian electronic patient data system). In LABSS, symptoms (yes/no) were collected by the study nurse through interviews during patient recruitment and recorded in a pre-designed consultation questionnaire. SOCCS and LABSS also collected blood samples, and DNA samples were genotyped using Illumina^®^ HumanHap300, HumanHap240S, and OmniExpressExome BeadChip 8v1 arrays. Genotype data quality control was performed following the method proposed by Anderson [23]. Untyped variants were imputed using the Michigan Imputation Server, which is based on 1000 genomes (from the European reference panel) [24].

### 2.2. Descriptive and Association Analysis

We performed a baseline summary for SOCCS and LABSS. The test of correlation and difference in variables between cases and controls in two studies were examined for statistical significance by using the *t*-test (continuous variables) and the Pearson χ2 test (categorical variables). Univariable and multivariable logistic regression models were fitted to test the associations between variables and CRC risk (factors with univariable *p* < 0.05 were included in the multivariable analysis).

### 2.3. Weighted Genetic Risk Scores

A weighted genetic risk score (wGRS) is defined as a weighted sum of dosages of risk alleles for k considered SNPs (g_i1_, …, g_ik_) for the n subjects (i = 1, …, n). The wGRS formula is: *GRS_i_ = w*_1_*g_i_*_1_
*+ … + w_k_g_ik_*. This means that, for each individual, the number of risk alleles dosages carried at each genetic variant SNP is summed, and it is weighted by its effect size. The effect size derived from the meta-GWAS for a SNP is referred to as the ‘weight’ (w_1_, …, w_k_). 

We used CRC genome-wide significant SNPs (*p* < 5 × 10^−8^; n = 202) from a recently published meta-GWAS study [25]. The meta-GWAS study investigated a total of 205 SNPs, and 202 SNPs effect sizes in European populations were reported (for SNPs list and their reported effect size, please see Appendix A). Of the 202 SNPs, 137 were genotyped in SOCCS and LABSS. We checked the remaining 65 SNPs for proxies. We found proxies for 26 SNPs (R^2^ > 0.5) and 39 SNPs (0.034 < R^2^ < 0.5). Therefore, we calculated three wGRSs to include 137 (genotyped SNPs), 163 (genotyped SNPs and 26 proxies with R^2^ > 0.5), and 202 (137 genotyped SNPs and 65 proxies) SNPs (Appendix A). We presented wGRS_202_ in the main text and the comparative assessment of model performance of wGRS_137_, wGRS_163_, wGRS_202_ is in Appendix A.

### 2.4. Model Development and Internal Validation

CRC prediction models’ development and validation were conducted and reported following the Transparent Reporting of a multivariable prediction model for Individual Prognosis or Diagnosis (TRIPOD) guideline [26] (Appendix A).

Models were developed with internal validation in the combined dataset with a total number of 2649 participants (CRC symptomatic cases = 1686, symptomatic controls = 963; Figure 1). The prediction outcome (Y) was defined as CRC (yes/no). Candidate predictors (X) included (i) continuous variables—age, BMI, and wGRS—as well as (ii) categorical variables—sex, family history, and symptoms (changes in bowel habits, rectal bleeding, weight loss, anaemia, and abdominal pain).

Each continuous variable (X) was modelled to test its association with the predicted outcome (Y) using two approaches: (i) linear analysis and (ii) restricted cubic splines (RCS). The continuous variables were then adjusted and incorporated into the full models C (linear) and E (RCS). The prediction performance, including overall accuracy (R^2^, brier score, AIC, BIC), discrimination (C-statistics), and calibration (*p*-value of Hosmer-Lemeshow test), were compared for the two approaches. The brier score (range: 0–1) quantifies the mean squared difference between the predicted probability and the observed outcome, with a lower score indicating a better prediction performance [27]. AIC and BIC are estimations concerning the sample prediction error, with a lower AIC or BIC value indicating a better model fit [28]. The decision on whether to use linear or RCS to adjust continuous variables in the final model was made by evaluating which method yielded better prediction performance.

After adjusting for the continuous variables (X), CRC risk prediction models were built (Figure 1). Two main strategies to develop the final models are predictor selection and full model [29]. A comparison of strengths and limitations of the methods is presented in Appendix A. Models A (baseline model + wGRS) and B (baseline model) were constructed based on LASSO regression algorithm to identify the *λ* (lambda) in response to the most parsimonious model where the cross-validation prediction error is within one standard error of the minimum [30]. The influential predictors selected by LASSO were incorporated into the prediction models. Models C (baseline model + wGRS) and D (baseline model) were built using all 10 variables collected in SOCCS and LABSS. These 10 variables were used as predictors in the 19 CRC prediction models previously developed (Appendix A), and, therefore, they were incorporated in models C and D, irrespective of their associations with the prediction outcome or influence on the model performance. In addition, we built prediction models F and G based on random forest regression [31,32], and the results were presented in Appendix A, Appendix A.

### 2.5. Model Prediction Performance

Models’ prediction performance was evaluated in terms of calibration and discrimination. Calibration, which measures the agreement between the model predicted probabilities (the risk rate of individuals with CRC) and the observed probabilities, was assessed using the Hosmer-Lemeshow (HL) goodness of fit test, with a *p* > 0.05 indicating good model calibration. Calibration curves were plotted to visualize the models’ calibrative power. Discrimination performance was examined through analysis of the area under the curve (AUC), which is also referred to as the C-statistic. The corrected C-statistics were calculated based on bootstrapping validation (1000 bootstraps resamples). The receiver operating characteristic (ROC) curve and the precision-recall curve (PRC) were plotted [33,34]. The continuous Net Reclassification Index (NRI) and Integrated Discrimination Index (IDI) were calculated after recalibration to compare models and assess the prediction increment [35]. An online nomogram for the final model was built using Shiny.apps.

### 2.6. Statistical Analysis

The LASSO regression was conducted using the ‘glmnet’ R package. Random forest regression was performed using the ‘randomForest’ R package. The HL test was constructed using the ‘hoslem.test’ function in the ‘ResourceSelection’ R package. The C-statistic was calculated using the “rcorr.cens” and “roc” functions in the ‘rms’ package. The online CRC risk prediction nomogram/calculator was constructed using the ‘DynNom’ and ‘rsconnect’ R packages. A two-sided *p*-value less than 0.05 was considered statistically significant. All analyses were performed using R, version 4.0.3 (R Foundation for Statistical Computing).

## 3. Results

### 3.1. Baseline Characteristics

The baseline characteristics of SOCCS (n = 1649) and LABSS (n = 1000) studies are summarized in Table 1. The distribution of each variable comparing symptomatic cases versus symptomatic controls in two studies is presented in Appendix A. There were no statistically significant differences between CRC symptomatic cases in SOCCS and LABSS with regards to wGRS_202_, age, sex, BMI, family history, and symptoms (*p* > 0.05). Comparing symptomatic cases (n = 1686) versus symptomatic controls (n = 963) in SOCCS and LABSS (Table 1), CRC symptomatic cases had a higher wGRS_202_, were older in age, and had a higher proportion of male patients, compared to symptomatic controls (*p* < 0.001). Cases had a lower BMI (*p* = 0.017). No statistically significant differences were found between symptomatic cases and controls for family history (*p* = 0.570). Regarding symptoms, the proportion of anaemia was significantly higher in CRC symptomatic cases (23.31%) than in the symptomatic control group (14.75%) [*p* < 0.001], while the proportions of changes in bowel habits (42.41%), weight loss (14.77%), and abdominal pain (19.69%) in CRC symptomatic cases were significantly lower compared to the symptomatic control group (changes in bowel habits: 74.87%, weight loss: 18.59%, abdominal pain: 43.93%) [*p* < 0.001]. Rectal bleeding was not statistically different between symptomatic cases and controls (*p* = 0.219).

In univariable analysis, statistically significant baseline factors for CRC risk included wGRS_202_, age, sex, BMI, and symptoms: changes in bowel habits, weight loss, anaemia, and abdominal pain (*p* < 0.05). Family history and rectal bleeding were not associated with CRC risk (*p* > 0.05). The above eight significant baseline factors were included in the multivariable analysis. Multivariable analysis demonstrated that (i) age (OR = 1.04, 95% CI: (1.03–1.05); *p* = 1.43 × 10^−28^), (ii) sex (male: OR = 1.44, 95% CI: (1.20–1.72); *p* = 7.11 × 10^−05^), (iii) wGRS_202_ (OR = 2.14, 95% CI: (1.74–2.64); *p* = 5.52 × 10^−13^), (iv) BMI (OR = 0.98, 95% CI: (0.97–1.00); *p* = 0.019), and (v) symptoms—changes in bowel habits (OR = 0.28, 95% CI: (0.23–0.34); *p* = 7.92 × 10^−37^), abdominal pain (OR = 0.51, 95% CI: (0.42–0.61); *p* = 8.48 × 10^−12^) remained independent predictors for CRC risk (Table 1).

### 3.2. Prediction Models of CRC Risk in Patients with Symptoms

Models A-D were developed with internal validation in SOCCS and LABSS to predict CRC risk in patients with symptoms (Figure 1).

#### 3.2.1. Continuous Variables Adjustment

The shape of the relationship between each continuous variable (age, BMI, and wGRS_202_) and the predicted outcome (CRC probability) is presented in Appendix A. Relationship figures showed steady increments in CRC probability for each year increase in age, decreasing BMI, and increasing wGRS_202_. The relationships between continuous variables and CRC were roughly linear in shape.

Continuous variables were then transformed by RCS, and we tested the hypothesis that the associations between continuous variables and the predicted outcome are not linear [36]. Spline functions with three, four, and five knots were created to fit each of these in the logistic regression model.

Appendix A demonstrated that R^2^, AIC, and BIC were the lowest using RCS with three knots, compared to four and five knots. There was no evidence of significant non-linear associations between age (nonlinear *p*-value = 0.105), BMI (nonlinear *p*-value = 0.587), wGRS_202_ (nonlinear *p*-value = 0.688), and CRC risk. The findings are consistent with Appendix A, showing that the relationships between age, BMI, wGRS, and CRC risk were linear in shape.

The continuous variables were adjusted and incorporated into the full model C (linear) and model E (RCS with three knots). Appendix A summarizes and compares the two models’ prediction performance. Model C had higher AIC, lower BIC, and higher corrected C-statistic compared to model E. Therefore, continuous variables (X) were adjusted in CRC prediction models, keeping age, BMI, and wGRS_202_ as continuous covariates in models. 

#### 3.2.2. Models’ Development and Validation

Each model’s predictors, intercept, coefficients, discrimination, and calibration estimates are presented in Table 2. Model formulas are presented in Appendix A.

CRC prediction models A, B, C, and D were evaluated, and they demonstrated good prediction performance. The summary of discrimination and calibration results for these models is as follows: Model A had a C-statistic of 0.767 (corrected 0.765) and a HL-test *p*-value of 0.024, while Model B had a C-statistic of 0.754 (corrected: 0.753) and a HL-test *p*-value of 0.711, as shown in Table 2 and Figure 2, Figure 3 and Figure 4. Model C had a C-statistic of 0.767 (corrected: 0.764) and a HL-*p* value of 0.018, while Model D had a C-statistic of 0.755 (corrected: 0.752) and a HL-*p* value of 0.428 (Table 2; Figure 5, Figure 6 and Figure 7). Precision recall curves, which visualize the relationship between precision (positive predictive value) and recall (sensitivity) to compare across models, were shown in Figure 4 and Figure 7.

Models A (parsimonious LASSO model) and C (full model) had better prediction performance, compared to baseline models B and D. The findings suggested incremental predictive value had been introduced by the addition of wGRS [Model A vs. B: NRI = 0.226 (0.149–0.335), IDI = 0.019 (0.013–0.024); Model C vs. D: NRI = 0.239 (0.154–0.340), IDI = 0.018 (0.013–0.023); *p* < 0.01]. There was no statistical difference in the predictive accuracy between models A and C (C-statistic increment = 0.001, *p* = 0.479). In addition, the sensitivity analysis found that there was no statistical difference in models for wGRS_137_, wGRS_163_, and wGRS_202_ predictive accuracy (Appendix A). Random forest models F (baseline model + wGRS) and G (baseline model), with 500 trees, were built, and the results were consistent with the findings in cross-assessment of models A/B and C/D (Appendix A). Model F had an out-of-bag (OOB) prediction error rate of 27.64%, compared to 27.37% for model G. Models that integrated wGRS in combination with demographic and clinical predictors had better performance than baseline models.

We developed an online CRC risk prediction nomogram/calculator A. This can be accessed through the following link: (https://crcpredictionmodel.shinyapps.io/dynnomapp/; accessed on 27 June 2023). The CRC risk for individuals can be calculated via inputting each patient’s information.

## 4. Discussion

### 4.1. Interpretation of Main Findings

Our study investigated the predictive value of demographic characteristics, a wGRS based on 202 CRC susceptibility SNPs, family history, and symptoms on CRC risk. The dedicated CRC prediction models were developed and internally validated for personalized cancer risk prediction for patients presenting with symptoms.

#### 4.1.1. Model Predictors

CRC risk prediction models A-D were constructed using a polygenic risk score, age, sex, BMI, family history, and symptoms to predict CRC risk in patients with symptoms.

In previous studies, a total of 19 CRC prediction models were developed [5,6,7,8,9,10,11,12,13,14,15,16,17,18,19,20,21,22]. The median number of predictors included in the models was ten (ranging from three to 16). An amount of 55 unique predictors were incorporated in at least one of the above 19 models (Appendix A). The 19models used predictors, such as demographic characteristics (age: in 16 models, 82.4%; sex in 11 models, 57.9%), lifestyle factors (smoking in four models, 21.1%; alcohol consumption in three models, 18.8%), biomarkers (haemoglobin in five models, 26.3%; CEA in two models, 10.53%), family history (in six models, 31.6%), and symptoms (rectal bleeding in 15 models, 78.9%; changes in bowel habits in 10 models, 52.6%; abdominal pain in nine models, 47.4%; weight loss in nine models, 47.4%; anaemia in five models, 26.3%).

The 10 candidate variables (except wGRS) in our study were all used as predictors in the previously developed 19 CRC prediction models. Our models’ findings were in line with these previous studies. It should be noted that family history data in SOCCS and LABSS studies was collected based on self-reported bowel cancer history, which was recorded in patient questionnaires and may be affected by recall bias. Furthermore, predictive value of symptoms as indicators for CRC is not well established. Previous studies argued that bowel symptoms correlate poorly with the presence of CRC [37]. They are also common in patients free from CRC risk, which implies they do not have good sensitivity for CRC [38]. Bowel symptoms are associated with CRC risk, but only for patients who have had the symptom at least weekly and for less than 12 months [5]. For symptoms that may be relevant, investigating the frequency and duration of symptoms is helpful. Data related to duration and frequency of bowel symptoms were unfortunately not collected in SOCCS, and thus we could not explore this in our study.

None of the 19 models incorporated genetic factors (neither individual SNPs nor a wGRS). To the best of our knowledge, this is the first study that developed and internally validated prediction models that included a wGRS in addition to demographic and clinical factors for CRC risk in patients with symptoms. Models A and C verified that the wGRS, including 202 CRC susceptibility SNPs, is the score with the best prediction performance, compared to baseline models B and D. The findings showed that the inclusion of the genetic predictor (wGRS) into the baseline model could improve CRC risk stratification. By comparison, previous studies were mainly focused on the predictive ability of genetic factors to capture the overall risk of CRC in the general population, not in symptomatic patients [39]. A recently published systematic review synthesized and evaluated a total of 33 CRC risk prediction models, which were developed by incorporating genetic predictors (SNPs or GRS) for the prediction of CRC risk in the general population [39] (Appendix A). An amount of 78.8% of the identified 33 CRC risk prediction models applied GRS, and the remaining 21.2% of them, incorporated SNPs as genetic predictors. The meta-analysis findings suggested no correlation between the number of SNPs and AUC improvement (*p* = 0.695). Furthermore, AUC improvement for the addition of genetic predictors to baseline models ranged from 0.010 to 0.084. The meta-analysis resulted in a pooled estimate of AUC improvement for genetic-enhanced prediction models compared with baseline models of 0.040 (95% CI: 0.035–0.045) [39]. 

These results are consistent with our finding of the polygenic risk score value in symptomatic patients. The integration of genetic predictors into classical CRC prediction models (baseline models) could improve the models’ prediction accuracy. There are several strengths for using genetic risk stratification in CRC. First, wGRS provides a measure of genetic susceptibility to CRC risk. Second, genetic predisposition to CRC remains relatively unchanged throughout life and affords the opportunity to provide long-term estimation of risk trajectories. Third, genetic risk stratification could improve CRC risk prediction in people who carry high-impact disease-causing genetic variants. Future application of genetic predictors holds significant promise and has the potential to enhance CRC risk prediction, assist clinical decision-making in precision therapeutics, and improve population-level screening [40]. Despite the potentials and benefits of using genetic predictors, there are risks and limitations of clinical use, which should be acknowledged. The first concern is to balance the cost and net benefit of using genetic predictors [40]. Genetic variants are not routinely collected in clinical practice, and it is not clear whether their predictive accuracy is better than for traditional risk factors, which can be more easily collected from routine patient records [39]. In addition, the standards and methods to incorporate genetic predictors in prediction models are constantly developing [41]. There has not been a unified standard, and this inconsistency becomes a major challenge during its clinical application. Another challenging aspect of using genetic predictors in clinical practice is to ensure that they are equally applicable to all ethnic groups [42]. The majority of current genetic variants data are from European populations, thus, GRS are primarily developed and validated in those of European descent [43]. This usually leads to a decrease in predictive accuracy when applied to non-European ancestries [44]. Lastly, it is important to validate genetic predictors’ feasibility in routine clinical practice [41]. It is suggested to evaluate the CRC genetic model’s clinical impact (e.g., cost-effectiveness) prior to implementation in the clinical setting [45].

#### 4.1.2. Model Prediction Performance, Validation, and Clinical Impact

CRC prediction models A, B, C, and D were found to have good predictive performance, surpassing the area under the ROC curves threshold of 0.7. Our models have the advantage of identifying symptomatic patients who have a higher probability of CRC among all patients. In addition, the calibration plots illustrated the acceptable agreement between the observed CRC probabilities and the predicted CRC probabilities. Due to a lack of external data, it was unfortunate that models A, B, C, and D could not be validated in the external population. Comparing LASSO model A and full model C, there was no statistical difference in the models’ predictive accuracy. It is critical to consider whether the model’s predictive accuracy increment is worth the additional time and cost to collect all the predictors. The parsimonious model A used five LASSO-selected influential predictors. LASSO approach could select the most influential predictors [46]. By comparison, the full model C used all the 10 predictors. In this study, the increased time and cost to collect the larger number of predictors for the full model C outweighed the increased predictive accuracy. It is important to balance model parsimony and accuracy [47]. From a practical perspective, the parsimonious model A is easier to interpret, generalize, and use in practice. In the current study, model A is preferred over model C. 

Compared to the previously published 19 risk prediction models, 13 (68.4%) models reported a median AUC value of 0.85 (ranged from 0.73 to 0.97), which indicated that these models had better discrimination ability. With regards to validation, 10 (52.6%) models did not undergo either internal or external validation; five (26.3%) models were internally validated; and three (15.8%) models were validated in external datasets. One model (5.3%) was developed with both internal and external validation. None of the 19 models performed clinical impact analysis. Although they perform at a level that is considered ‘clinically acceptable’ with a C-statistic >0.7, however, these models have not yet been applied in clinical practice.

### 4.2. Strengths and Limitations

The main strength of this study is that CRC prediction models were developed with internal validation to alleviate the models’ overfitting and optimism. Models incorporated both influential genetic and non-genetic predictors to increase the models’ prediction performance, which were validated to have good calibration and discrimination.

However, the following potential limitations should be considered. (1) This risk prediction modelling study was based on a small sample size and may not be sufficiently representative of the population. Furthermore, due to the small sample size, we did not develop risk prediction models for CRC risk in males and females separately or in different CRC cancer sites. (2) The majority of CRC cases came from SOCCS (97.81%), and all controls, were from LABSS. The different variable collection methods in SOCCS (GP e-referrals) and LABSS (questionnaire) could bias the study’s results. For GP e-referrals, it is possible that not all the symptoms would be accurately recorded by GPs. By comparison, for LABSS, patients were asked whether they had presented the symptoms (those were variables of interest and were designed to be collected in the questionnaire), and, therefore, they were more likely to recall a greater number of symptoms. (3) Previous systematic reviews found that biomarkers (e.g., haemoglobin, CEA, qFIT result), lifestyle (e.g., vitamin D) variables, and bowel symptoms (e.g., rectal mass, abdominal mass) are associated with CRC risk [48,49]. However, these predictors were not collected in SOCCS and LABSS studies and could not be employed in the developed CRC prediction models. (4) The prediction performance of using genetic predictors may vary, depending on the SNPs included (whether they are high-risk susceptibility), SNPs weight estimates from a meta-GWAS dataset, and the specific computational method used for GRS construction [39]. We included a list of genome-wide CRC significant SNPs (*p* < 5 × 10^−8^) from the most recently published meta-GWAS study [25]. However, 8.43% of the meta-GWAS participants were SOCCS participants. Thus, this could overestimate our wGRS when we used their SNPs’ coefficients for external weight. Another limitation is that current genetic variants are from European populations, which usually leads to a decrease in predictive accuracy when applied to non-European ancestries [50]. (5) Internal validation cannot address selection bias with recruitment, or measurement errors, as validation is performed within the study population [51]. (6) The C-statistic, HL goodness of fit test, and calibration plots were employed to examine model performance (discrimination and calibration). These metrics have their own limitations. The C-statistic does not have a clear interpretation when assessing the incremental value after adding a new predictor [52]. The HL test might lack statistical power to detect overfitting, it is sensitive to the sample size, and it provides no information on the direction or magnitude of miscalibration [53]. The calibration plot cannot provide quantitative assessment of model calibration [54]. (7) The developed CRC risk prediction models have not been externally validated due to lack of data. Validation studies of large sample size may be considered in the future.

### 4.3. Clinical Implications and Future Research

CRC prediction models have the benefit of providing disease risk assessment to identify patients, whilst also supporting clinical decision-making about risk-tailored, personalised clinical care [55]. This eventually could improve patients’ health outcomes and the cost-effectiveness of care [38]. Despite their benefits, CRC prediction models in front-line clinical practice remain under-utilized. There are risks and limitations of CRC prediction models in clinical use. The first concern is associated with prediction accuracy. Incorrect CRC prediction models might prioritize the wrong patients for further screening, interventions, and clinical treatments [56]. In addition, two studies conducted interviews/focus groups and surveys to investigate attitudes regarding the use of CRC prediction models among GPs and to identify barriers to their clinical use [57,58]. The findings indicate that clinicians may interpret symptoms inconsistently which would lead to inaccurate and unreliable CRC risk assessment. Therefore, future application of genetic predictors holds significant promise and has the potential to enhance CRC risk prediction.

## 5. Conclusions

CRC prediction models were developed with internal validation for personalized cancer risk prediction for patients presenting with symptoms. The integration of genetic architecture into the CRC classical prediction model could improve prediction performance. This could be helpful to identify a subpopulation among the symptomatic population with higher CRC risk due to genetic susceptibility. The findings merit further investigation through model external validation and model clinical impact.

## Figures and Tables

**Figure 1 jpm-13-01065-f001:**
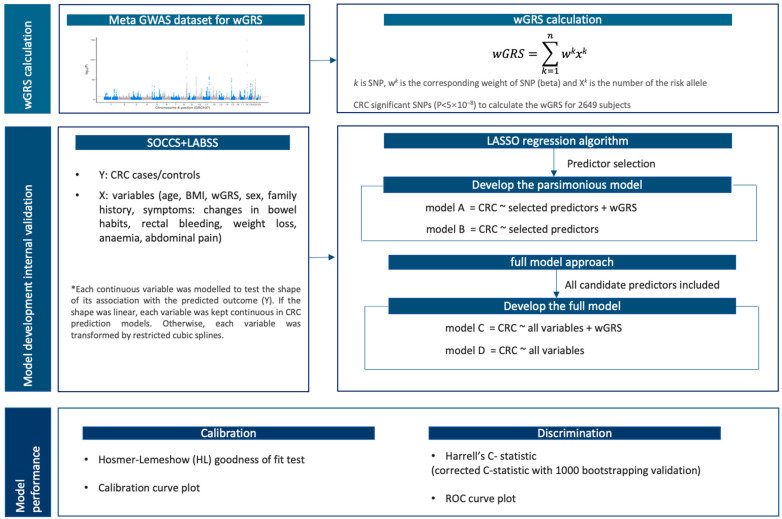
The CRC prediction models’ construction and internal validation.

**Figure 2 jpm-13-01065-f002:**
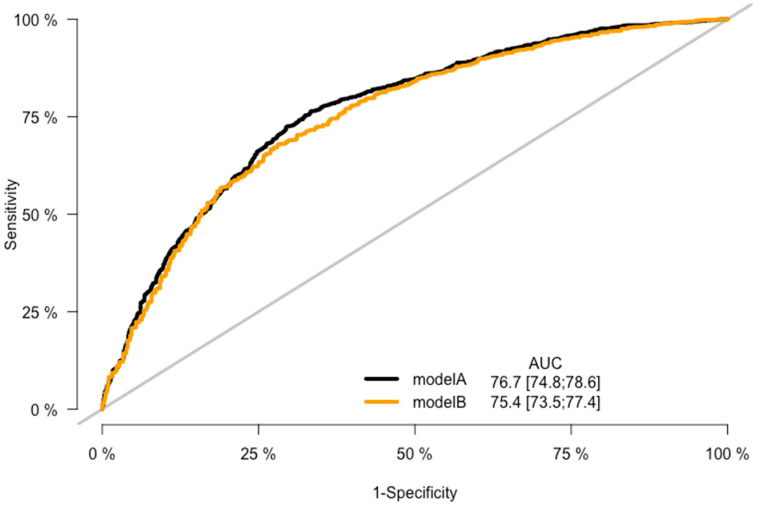
ROC curves—the model A and model B comparison.

**Figure 3 jpm-13-01065-f003:**
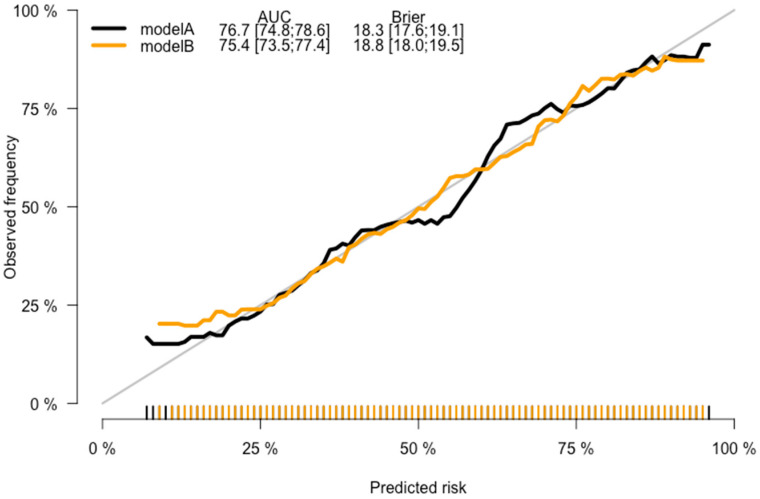
Calibration curves—the model A and model B comparison.

**Figure 4 jpm-13-01065-f004:**
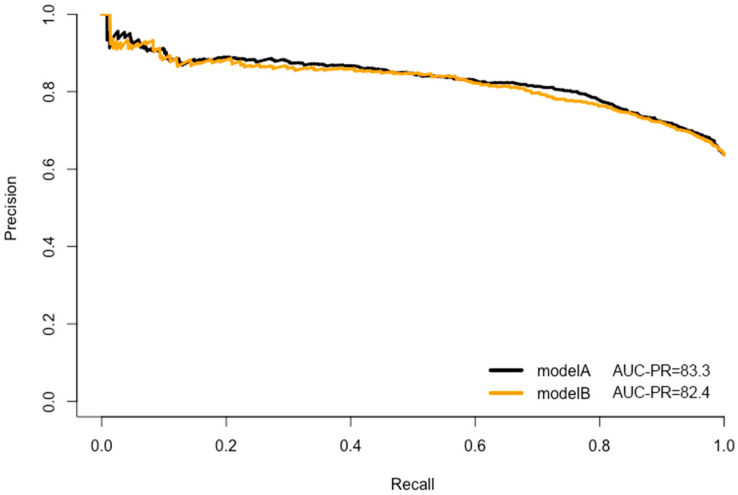
Precision recall curves—the model A and model B comparison.

**Figure 5 jpm-13-01065-f005:**
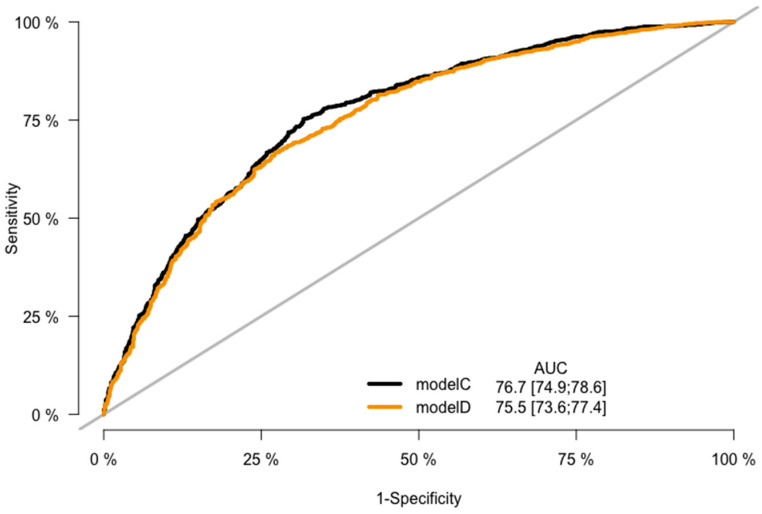
ROC curves—the model C and model D comparison.

**Figure 6 jpm-13-01065-f006:**
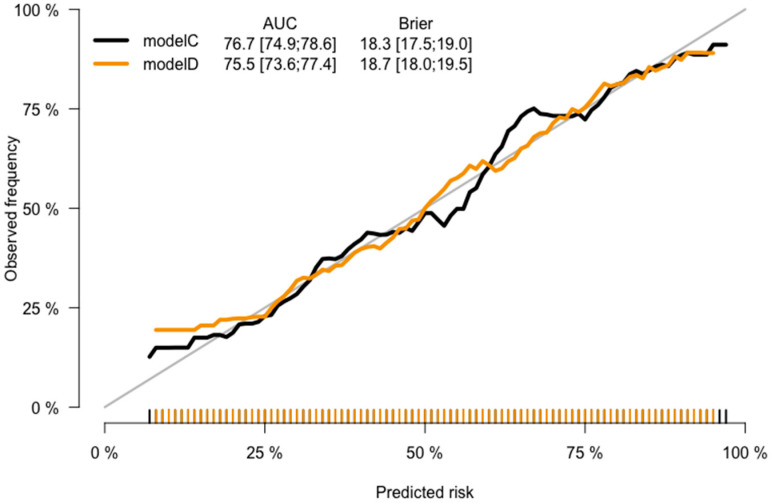
Calibration curves—the model C and model D comparison.

**Figure 7 jpm-13-01065-f007:**
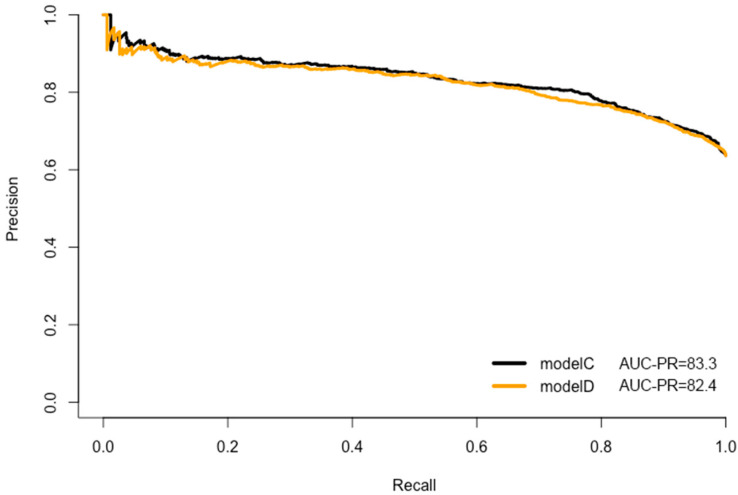
Precision recall curves—the model C and model D comparison.

**Table 1 jpm-13-01065-t001:** The univariable and multivariable logistic regression models of CRC risk.

	SOCCS + LABSS (N = 2649)				Univariable Analysis			Multivariable Analysis		
	Cases (n = 1686)	Controls (n = 963)	Total (N = 2649)	*p*-Value	OR	95% CI	*p*-Value	OR	95% CI	*p*-Value
wGRS_202_ †	0.11 (−0.19–0.42)	−0.03 (−0.34–0.26)	0.06 (−0.24–0.37)	3.36 × 10^−16^	2.14	1.77–2.58	1.88 × 10^−15^	2.14	1.74–2.64	5.52 × 10^−13^
Age †	68.01 (59.32–75.36)	60.00 (51.00–70.00)	65.42 (56.00–73.50)	<2.2 × 10^−16^	1.05	1.04–1.05	3.61 × 10^−42^	1.04	1.03–1.05	1.43 × 10^−28^
Sex										
Female	730 (43.30%)	537 (55.76%)	1267 (47.83%)	8.38 × 10^−10^	1 *			1 *		
Male	956 (56.70%)	426 (44.24%)	1382 (52.17%)		1.65	1.41–1.94	7.35 × 10^−10^	1.44	1.20–1.72	7.11 × 10^−5^
BMI †	26.11 (23.39–29.91)	26.64 (23.50–30.47)	26.35 (23.44–30.11)	0.017	0.98	0.97–1.00	0.016	0.98	0.97–1.00	0.019
Family history										
No	1418 (84.10%)	801 (83.18%)	2219 (83.77%)	0.570	1 *					
Yes	268 (15.90%)	162 (16.82%)	430 (16.23%)		0.93	0.75–1.16	0.534			
Symptoms										
Changes in bowel habits										
No	971 (57.59%)	242 (25.13%)	1213 (45.79%)	<2.2 × 10^−16^	1 *			1 *		
Yes	715 (42.41%)	721 (74.87%)	1436 (54.21%)		0.25	0.21–0.29	2.12 × 10^−55^	0.28	0.23–0.34	7.92 × 10^−37^
Rectal bleeding										
No	1130 (67.02%)	622 (64.59%)	1752 (66.14%)	0.219	1 *					
Yes	556 (32.98%)	341 (35.41%)	897 (33.86%)		0.90	0.76–1.06	0.203			
Weight loss										
No	1437 (85.23%)	784 (81.41%)	2221 (83.84%)	0.012	1 *			1 *		
Yes	249 (14.77%)	179 (18.59%)	428 (16.16%)		0.76	0.61–0.94	0.010	0.99	0.78–1.26	0.910
Anaemia										
No	1293 (76.69%)	821 (85.25%)	2114 (79.80%)	1.69 × 10^−07^	1 *			1 *		
Yes	393 (23.31%)	142 (14.75%)	535 (20.20%)		1.76	1.42–2.17	1.61 × 10^−07^	0.94	0.73–1.20	0.619
Abdominal pain										
No	1354 (80.31%)	540 (56.07%)	1894 (71.50%)	<2.2 × 10^−16^	1 *			1 *		
Yes	332 (19.69%)	423 (43.93%)	755 (28.50%)		0.31	0.26–0.37	1.03 × 10^−38^	0.51	0.42–0.61	8.48 × 10^−12^

SOCCS: the Study of Colorectal Cancer in Scotland; LABSS: and the Lothian Bowel Symptoms Study; OR: odds ratio; CI: confidence interval. * Reference group. Only significant factors (univariable *p* < 0.05) were included in the multivariable analysis. *p*-value for *t*-test or x^2^ test. † Median and quartiles in parenthesis.

**Table 2 jpm-13-01065-t002:** A summary of CRC prediction models A–D.

Model	Method	Case	Control	λ	Intercept	Predictors	Coefficient	OR (95% CI)	*p*-Value	R^2^	Brier	AIC	BIC	C-Statistic	Corrected C-Statistic	AUC-PR	HL *p*-Value
Model A	LASSO	1686	963	0.0257	−1.3030	wGRS_202_	0.7612	2.14 (1.74–2.64)	5.31 × 10^−13^	0.266	0.183	2911.234	2946.526	0.767 (0.748–0.786)	0.765 (1000 bootstrap)	0.8325	0.024
						Age	0.0410	1.04 (1.03–1.05)	3.53 × 10^−29^								
						Sex	0.3611	1.43 (1.20–1.72)	7.19 × 10^−5^								
						Changes in bowel habits	−1.2411	0.29 (0.24–0.35)	8.06 × 10^−29^								
						Abdominal pain	−0.6784	0.51 (0.42–0.62)	7.65 × 10^−12^								
Model B	LASSO	1686	963	0.0310	−1.2124	Age	0.0401	1.04 (1.03–1.05)	1.06 × 10^−28^	0.244	0.188	2962.840	2992.25	0.754 (0.735–0.774)	0.753 (1000 bootstrap)	0.8243	0.711
						Sex	0.3690	1.45 (1.21–1.73)	4.09 × 10^−5^								
						Changes in bowel habits	−1.2411	0.29 (0.24–0.35)	1.34 × 10^−39^								
						Abdominal pain	−0.7020	0.50 (0.41–0.60)	7.77 × 10^−13^								
Model C	Full model	1686	963	NA	−0.7679	wGRS_202_	0.7603	2.14 (1.74–2.64)	6.91 × 10^−13^	0.269	0.183	2915.181	2979.883	0.767 (0.749–0.786)	0.764 (1000 bootstrap)	0.8334	0.018
						Age	0.0410	1.04 (1.03–1.05)	2.65 × 10^−28^								
						Sex	0.3631	1.44 (1.20–1.72)	7.05 × 10^−5^								
						BMI	−0.0195	0.98 (0.96–1.00)	0.0187								
						Family history	−0.0024	1.00 (0.78–1.27)	0.9846								
						Changes in bowel habits	−1.2616	0.28 (0.23–0.34)	7.68 × 10^−37^								
						Rectal bleeding	0.0402	1.04 (0.86–1.27)	0.6858								
						Weight loss	−0.0112	0.99 (0.78–1.26)	0.9278								
						Anaemia	−0.0531	0.95 (0.74–1.22)	0.6785								
						Abdominal pain	−0.6786	0.51 (0.42–0.63)	1.55 × 10^−11^								
Model D	Full model	1686	963	NA	−0.7170	Age	0.0404	1.04 (1.03–1.05)	4.12 × 10^−28^	0.247	0.187	2966.240	3025.059	0.755 (0.736–0.775)	0.752 (1000 bootstrap)	0.8240	0.428
						Sex	0.3714	1.45 (1.21–1.73)	3.94 × 10^−5^								
						BMI	−0.0191	0.98 (0.97–1.00)	0.0200								
						Family history	−0.0349	1.04 (0.82–1.32)	0.7738								
						Changes in bowel habits	−1.2667	0.28 (0.23–0.34)	7.07 × 10^−38^								
						Rectal bleeding	0.0734	1.08 (0.89–1.31)	0.4553								
						Weight loss	−0.0661	0.99 (0.78–1.27)	0.9655								
						Anaemia	−0.6999	0.94 (0.73–1.20)	0.6021								
						Abdominal pain	−0.6786	0.50 (0.41–0.60)	2.03 × 10^−12^								

AIC: Akaike’s Information Criteria; AUC-PR: area under the precision recall curve; BIC: Bayesian information criteria; CI: confidence interval; HL: Hosmer-Lemeshow; OR: odds ratio.

## Data Availability

The data presented in this study are available upon reasonable request to the corresponding author.

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
