# Peer review of "Development and Validation of Risk Prediction Models for Colorectal Cancer in Patients with Symptoms"

_jpm, 2023, doi:10.3390/jpm13071065_

Round 1

Reviewer 1 Report

Any major comments and suggestions to Authors.

An interesting methodology for modeling and forecasting symptoms due to colon cancer was chosen. The article is original in its subject matter. Publications on this topic have not agreed.

Author Response

Response:

We appreciate the reviewer’s insightful feedback and acknowledgment of our article. We thank the reviewer’s recognition of the chosen methodology for modeling and forecasting symptoms associated with CRC risk in patients with symptoms.

Reviewer 2 Report

The authors develop and validate predictive models that combine demographics, clinical features, and weighted genetic risk scores (wGRS). The authors found that models combining wGRS with demographic and clinical predictors statistically significantly improved predictive performance. And expect this to improve the risk profile of the CRC. There are a few issues to focus on here.

1.        From Figure 2 to Figure 4, the difference in AUC between the four models is actually not much. Are the authors considering using other metrics, such as the Precision-Recall curve?

2.        For figure 6, please do not use online CRC risk prediction nomogram/calculator. Would you please redraw a figure.

3.        From the analysis of indicators such as R2 in Table 2, the difference between the four models is relatively small. Whether the authors consider using other non-LASSO models, such as random forests, support vector machines, for modeling?

No

Author Response

Response:

Thank reviewer for your thoughtful review of our article. We have carefully considered the points the reviewer raised, and we would like to address them as follows:

  1. Regarding the difference in AUC between the four models (Figure 2 to Figure 4), we agree that AUC (difference is relatively small) alone may not provide a comprehensive evaluation of the models' performance. We appreciate the reviewer’s suggestion to consider additional metrics such as the Precision-Recall curve, which can provide insights into the trade-off between precision and recall for different classification thresholds. We have added Figures: Precision recall curves-model A and model B comparison/ model C and model D comparison. (Pages: 9-13) Models A and C that integrated wGRS in combination with demographic and clinical predictors had higher AUCpr than baseline models B and D. The results were in line with ROC curves.

  1. We thank the reviewer for this comment. We have removed figure 6 (online CRC risk prediction nomogram/calculator) (Page: 14).

  1. We appreciate the reviewer’s suggestion to explore non-LASSO models, such as random forests and support vector machines for modeling. While our current study focused on the LASSO model (predictor selection) and full model, we acknowledge the potential benefits of considering alternative modeling methods. We have built Model F (baseline model + wGRS) and G (baseline model) with 500 trees based on random forest regression. (Supplementary Table 12; Figures 11-13) Model F had an out-of-bag (OOB) prediction error rate of 27.64%, compared to 27.37% for model G. Model F had higher C-statistics, sensitivity, specificity, positive predictive value, and negative predictive value than model G. The results were consistent with LASSO model and full model. We found that incremental predictive value had been introduced by the addition of wGRS.

Once again, we appreciate the thorough review and valuable suggestions. The reviewer’s feedback will significantly enhance the quality and clarity of our manuscript.
